# The Role of Sex in Acute and Chronic Liver Damage

**DOI:** 10.3390/ijms231810654

**Published:** 2022-09-13

**Authors:** Katia Sayaf, Daniela Gabbia, Francesco Paolo Russo, Sara De Martin

**Affiliations:** 1Department of Surgery, Oncology and Gastroenterology, University of Padova, 35131 Padova, Italy; 2Department of Pharmaceutical and Pharmacological Sciences, University of Padova, 35131 Padova, Italy; 3Gastroenterology and Multivisceral Transplant Units, Azienda Ospedale—Università di Padova, 35131 Padova, Italy

**Keywords:** sex, chronic liver disease, acute liver failure, liver regeneration

## Abstract

Acute and chronic hepatic damages are caused by xenobiotics or different diseases affecting the liver, characterized by different etiologies and pathological features. It has been demonstrated extensively that liver damage progresses differently in men and women, and some chronic liver diseases show a more favorable prognosis in women than in men. This review aims to update the most recent advances in the comprehension of the molecular basis of the sex difference observed in both acute and chronic liver damage. With this purpose, we report experimental studies on animal models and clinical observations investigating both acute liver failure, e.g., drug-induced liver injury (DILI), and chronic liver diseases, e.g., viral hepatitis, alcoholic liver disease (ALD), non-alcoholic fatty liver disease (NAFLD), autoimmune liver diseases, and hepatocellular carcinoma (HCC).

## 1. Introduction

The liver plays a central role in many physiological processes, including metabolism, energy and lipid storage, and xenobiotic detoxification. The liver is a highly sexual dimorphic organ, accounting for at least 72% of sexually differentiated genes [1]. This sex difference is reflected in the different susceptibility, progression, and outcomes of many acute and chronic liver diseases in females and males. The aim of this review is to summarize the recent findings on the sex difference in the development of acute and chronic liver damages, and the role of the sex-specific pathways responsible for the sex dimorphism, with a particular focus on the regulation exploited by sex hormones on their nuclear receptors. 

Sex steroid hormones finely regulate a variety of pathophysiological processes in the liver, including lipid metabolism, inflammation and fibrogenesis (Figure 1). Their deregulation is implicated in the development of hepatocellular damages of different severities. Sex steroid hormones are divided into three classes, i.e., androgens, estrogens, and progestins. Androgens and testosterone, which is the most biologically relevant androgenic hormone, are considered male-specific sex hormones, induce masculinizing effects, and regulate male sexual behavior. On the other hand, estrogens and 17β-estradiol (E2) are female-specific sex hormones, and are devoted to regulating female reproductive physiology and behaviors. Progestins, among which the most relevant human hormone is progesterone, play essential roles in the regulation of the reproductive system in both females and males. In females, progesterone regulates the maintenance of pregnancy and the menstrual cycle, and prepares mammary glands for lactation and breastfeeding. In males, progesterone is involved in the regulation of testosterone synthesis, spermiogenesis, and sperm capacitation. The peculiar features of different sex hormones are displayed in a different manner in the two sexes due to their sexual-dependent circulating levels, as all sex hormones are present in both males and females.

Sex steroid hormones can mediate peculiar physiologic effects by binding to specific receptors. Most of them are nuclear receptors (NRs) that, once activated by the sex hormone binding, dimerize, and translocate to the nucleus, where they orchestrate the transcription of target genes, helped by coregulators [2]. In addition, the existence of non-nuclear steroid receptors has been reported. Upon ligand binding, they mediate more rapid, non-nuclear or non-genomic responses [3]. In addition, the subcellular distribution of sex hormone receptors could be responsible for the multiple physiological effects exerted in different tissues.

The aim of this review is to summarize the most recent advances in the comprehension of the molecular basis of the sex differences observed during acute liver failure, e.g., DILI, and chronic liver diseases, e.g., viral hepatitis, ALD, NAFLD, autoimmune liver diseases, and HCC, describing the relevant findings obtained in both preclinical experimental studies and clinical observations.

## 2. Sex Hormone Signaling and Hepatic Metabolism

### 2.1. Estrogens

Estrogen receptors (ERs) include the NR, ER𝛼 and ER𝛽; the membrane-bound G protein-coupled estrogen receptor (GPR30 or GPER); and the membrane ER𝛼 and ER𝛽 variants [4]. In human and murine livers of both sexes, these receptors are constitutively expressed, although at lower levels than in reproductive organs [5,6]. In hepatocytes, ER𝛼 is the most represented subtype, and is involved in the regulation of lipogenesis. Three estrogens activate its signaling, i.e., E1 (estrogen), E2, and E3 (estriol), but dietary amino acids (AAs) can also activate hepatic ERα, which concurs to the regulation of the estrous cycle [7]. Thus, it has been hypothesized that the difference in the hepatic metabolism of these dietary AAs observed in the two sexes could be directly related to the female-specific role of hepatic ERα in the regulation of hepatic energy production, to support reproductive functions. Moreover, it has been suggested that hepatic ER𝛼 could sense the fluctuation of circulating E2 levels during life and the estrous cycle in response to reproductive cues, thus modulating hepatic metabolism to adapt the energy requirements in each stage. This isoform is likely to be responsible for the metabolically protective effect of estrogens toward different hepatic injuries [8,9]. Estrogens also play a role in hepatic glucose homeostasis and insulin clearance, being able to decrease gluconeogenesis and increase hepatic glycogen synthesis and storage, thus regulating glucose plasma levels [10,11]. As an example, E3 is more effective than E2 in decreasing the post-prandial glucose peak through the downregulation of the glucose transporter GLUT2 during pregnancy.

### 2.2. Androgens

Androgens—i.e., testosterone, dihydrotestosterone (DHT), dehydroepiandrosterone (DHEA), androstenedione, androstenediol, and androsterone—play an important role in the orchestration of hepatic metabolic homeostasis. Testosterone and DHT can bind directly to their receptor, whereas the other four hormones are considered to be pro-androgens, and should be converted to testosterone to exploit their androgenic effects. In both sexes, androgens rapidly rise during puberty reaching a peak, after which their concentrations gradually decline. In males, androgen levels are about 20 times higher than in females [12,13]. Androgens could bind to nuclear androgen receptors (ARs), thus regulating the transcription of many genes; they also exert a non-genomic effect, acting on the MAPK (mitogen-activated protein kinase) and the PI3K/AKT (phosphatidylinositol-3-kinase/AKT) pathways, or binding membrane-associated ARs or other membrane receptors, such as the epidermal growth factor receptor (EGFR) [14,15,16].

Although it is known that testosterone-binding to ARs promotes the physiological transcription of the insulin receptor in males, its effect on insulin signaling remains unclear, and conflicting evidence is reported regarding the fact that testosterone supplementation is effective or deleterious in the improvement of insulin sensitivity [17,18]. In a mouse model of type 2 diabetes (T2D), testosterone supplementation worsens hepatic insulin resistance in males [19]. Testosterone is also able to regulate the physiological synthesis of glycogen and decrease glucose uptake by controlling the transcription of the glucose transporter GLUT2 [20,21]. In diabetes and coronary artery disease, testosterone can exert an anti-inflammatory effect by acting on the adipose tissue, by increasing the circulating levels of anti-inflammatory cytokines, such as IL10, while decreasing pro-inflammatory mediators, such as, e.g., IL-1β, IL-6, and TNF-α [22]. Moreover, in male rats with liver steatosis, androgens have been demonstrated to reduce the risk of a progression to non-alcoholic steatohepatitis (NASH) by downregulating the proinflammatory cytokines TNF-α and IL-6 [23]. In females, testosterone impairs hepatic glucose metabolism, thus increasing the risk of developing impaired glucose tolerance and T2D [24,25,26]. In a mouse model of acute liver injury, a delayed resolution of necrotic damage and a higher expression of proinflammatory cytokines were observed in male mice, along with a slower recruitment of inflammatory monocytes, characterized by the expression of AR. Interestingly, their recruitment was modulated by the AR antagonist flutamide.

### 2.3. Progestins

In the liver, progestins do not bind the classical nuclear receptors but could exert their metabolic effect through membrane progesterone receptors or ARs [27,28]. The role of progestins in the development of some liver diseases, such as NAFLD, is still to be completely elucidated, but some evidence has linked increased circulating levels of progesterone to prediabetes and the onset of insulin resistance [29].

## 3. Acute Liver Injury

Acute liver failure (ALF) is an unexpected and rare disease, and is less common than chronic liver diseases [30]. Its pathophysiological mechanisms are still controversial. The incidence of ALF is about 1 to 2000 cases per million per year, including people with no preexisting diseases which manifest severe liver cell injury, such as toxic necrosis, apoptosis, or immune-mediated damages [31,32]. ALF cases have been reported as the result of multiple causes, including hepatitis A, B, E, ischemia, DILI, and autoimmunity. Being a potentially reversible condition, understanding the right etiology is fundamental for the choice of the correct treatment [31]. ALF is considered to be an unpredictable condition in most cases, and its diagnosis is often difficult during the early stages of the disease, leading to an underestimation of the situation by the patients [33]. The decline of liver function is unexpected, and is characterized by a rapid onset, with severe manifestations such as hepatic encephalopathy, acute kidney injury, cardiovascular problems with coagulopathy (international normalized ratio [INR] > 5), cerebral edema, and sepsis, leading to a high mortality rate [31,33]. To date, the only effective treatment for the advanced stages of ALF characterized by irreversible liver damage is liver transplantation (occurring in about 25% of ALF patients), although lethal complications are still common within the 3 months following the transplantation [33].

The role of sex in influencing the pathophysiological mechanisms and outcomes of ALF is far from completely understood. However, a role of sex hormones has been proposed, as well as sex-related modifications of gene transcription and consequent metabolic disorders [34,35]. Generally, about 67–70% of ALF patients are women, especially in the case of DILI, mirroring the typical susceptibility of women to adverse drug reactions (ADRs) [36]. In a mouse model of acute liver injury, a delayed resolution of necrotic damage and a higher expression of proinflammatory cytokines were observed in male mice, along with a slower recruitment of inflammatory monocytes, characterized by the expression of AR. Interestingly, their recruitment was modulated by the AR antagonist flutamide [37]. 

### 3.1. Drug-Induced Liver Injury 

It is reported that about 5–10% of people undergoing pharmacological treatments manifest ADRs, which can be—in some cases—severe, leading to hospitalization, permanent injury, or even death in the worst case [38]. For several drugs, sex-related differences have been reported in the frequency of ADRs, and generally women experience more severe and frequent adverse events than men, even though some bias could be related to the fact that in many clinical trials men used to be better represented than women. In fact, in the past, females were not equally enrolled in clinical trials, tending to be excluded from phase 1 and 2 [38,39,40].

DILI is often an idiosyncratic ADR, and is not related to the mechanism of action of the drug, independent of the dose [38], and is usually caused by a toxic metabolite produced by the numerous drug-metabolizing enzymes expressed in the liver [41]. Intrinsic DILI can also occur when it induces a liver injury in a dose-dependent manner [42]. DILI represents the most common cause of ALF, and antibiotics represent the cause of 50% of DILI cases. Its incidence is about 14–19 events per 100,000 inhabitants, with a prevalence of about 60–70% in women [43]. The risk of experiencing DILI is related to multiple factors, including sex-related differences in drug bioavailability, metabolism, and excretion, as well as immune-related causes linked to genetic backgrounds [31,34]. To date, convincing results have demonstrated an association between human leukocyte antigen (HLA)-A*33:01 and the DILI caused by fenofibrate, ticlopidine, and sertraline. Interestingly, Nicoletti and colleagues performed a genome-wide association (GWAS) study in live injury patients, finding some polymorphisms that could be associated with DILI. They concluded that a haplotype comprising the three correlated alleles of HLA-A*33:01, B*14:02, C*08:02 might be a risk factor for DILI, more so than the presence of the HLA-A*33:01 alone. Many more other risk alleles have been associated to DILI, such as HLA-DRB1*0701 and HLA-B*5701 for flucloxacillin-related DILI and further complications, including cutaneous hypersensitivity [44]. For this reason, genetic testing for the identification of HLA alleles might be a useful tool for prevention, also confirming the importance of adaptative immune responses in the pathophysiology of DILI. In fact, one of the hypothetical mechanisms proposed for DILI pathogenesis involves the formation of irreversible adducts between the drug, or its metabolites, and endogenous proteins expressed by hepatic cells [45]. The resultant complex is then phagocytosed by antigen-presenting cells (APCs), and is immediately exposed on their surface with major histocompatibility complex (MHC II) molecules to allow the activation of T cells. This process triggers the immune system for immune-mediated injuries in the liver [46]. Focusing on sex, factors which help us to understand why DILI predominantly occurs in women include the effects of sex hormones, differences in body composition, basal metabolic activity, and pregnancy. Sex hormones might also increase the risk of dose-dependent DILI for pharmacokinetic reasons, as progesterone contraceptives or substitutive hormone therapies may be responsible for drug–drug interactions [41]. Furthermore, it is widely known that estrogens regulate gastric motility, delaying stomach emptying and modulating the bioavailability of highly diffusible drugs [38,47]. In women, hormonal fluctuations during menstruation and pregnancy are responsible for decreased levels of plasma albumin, leading to increased levels of unbound drugs [38,48]. Pregnancy-related hormones (PRH) such as cortisol, estrogens, placental growth hormones and prolactin affect hepatic drug metabolism. Even if the underlying mechanisms are still unclear, evidence has demonstrated that PRH increase the levels of cytochrome P450 (CYP) 2D6 and 3A4, while the activity of CYP1A2 and P-glycoprotein (P-gp) decrease in this condition [47,49]. CYP3A4 and CYP2B6 are physiologically more abundant in women than in men due to higher concentrations of serum growth hormone (GH) [50]. Indeed, this hormone is strictly connected with human hepatic enzymes, as it regulates their gene transcription and transduction [38]. For this reason, variations in CYP isoform expression and the higher renal clearance in males could represent the primary cause of DILI in women [34]. 

The physicochemical characteristics of drugs might influence their bioavailability, as high octanol–water partition coefficient logPs (typical of lipophilic drugs) are frequently associated with the onset of DILI, enhancing the drug uptake by the hepatocytes. As stated before, DILI can occur because of the formation of active metabolites or because of oxidative stress in the hepatic parenchyma, as reactive oxygen species could damage DNA by altering the respiratory chain of mitochondria, and could lead to hepatocyte death [43]. In this regard, it has been demonstrated that sex hormones play a role in the regulation of hepatic pro-inflammatory cytokines. Cho and colleagues demonstrated that estrogens enhance the production of interleukin-6 (IL-6) in mice, causing the suppression of regulatory T-cells (Tregs), thereby inhibiting their anti-inflammatory action in a DILI-induced murine model [51,52]. A study analyzing the immune phenotype of DILI patients demonstrated a dimorphic immune response, as both males and females were characterized by a massive recruitment of monocytes to the liver, but only in males was this recruitment sustained by a turnover of immature monocytes [37].

### 3.2. ALF-Associated Comorbidities

#### 3.2.1. Ischemic Hepatitis

Ischemic hepatitis (IH), or “shock liver” is a condition of insufficient hepatic blood flow and/or oxygen content in the hepatocytes, as a consequence of cardiac acute diseases such as cardiac arrest, hypotension or cardiopulmonary collapse [53,54]. IH is an extremely common cause of ALF, and is usually associated with an increase of liver enzymes, including aspartate aminotransferase (AST) and alanine aminotransferase (ALT) [54,55,56]. Oxygen delivery is fundamental for tissue homeostasis. When IH occurs, the low pressure of oxygen in the arteries leads to a massive centrilobular necrosis of hepatocytes, contributing to liver failure [57]. Most of the HI-associated ALFs are diagnosed in the intensive care unit (ICU). The incidence is 1–2.5%, with no significant evidence for sex dimorphism [57,58]. However, a prospective study from Taylor and colleagues found that 63% of IH patients developing ALF were females, while only 37% were males [59]. 

#### 3.2.2. Ischemia Reperfusion Injury 

Hepatic ischemia reperfusion injury (IRI), which occurs during or after liver transplantation, is the injury occurring when the blood flow to the liver is first dismissed and then repristinated [60,61]. Patients with chronic liver diseases awaiting liver transplantation are susceptible to IRI because they have a compromised coagulation, which is characterized by high levels of pro-thrombotic factors and hypoalbuminemia [61]. The process underlying IRI is complex and involves different molecular pathways such as ischemia-induced cellular damage and reperfusion-induced inflammation [62]. During ischemia, IRI contributes to cell damage by promoting necrosis, while the cellular debris from dead hepatocytes could function as a damage-associated molecular pattern (DAMPs) [60,62]. During the reperfusion period, DAMPS, pro-inflammatory cytokines of and reactive oxygen species (ROS) activate macrophages and neutrophils that worsen hepatic injury, leading to ALF [61,63]. One of the risk factors for an unsuccessful liver transplantation includes the sex mismatch of donor/recipient, but the underlying mechanisms for this discrepancy have to be further investigated [64]. To date, no clinical data are available regarding sex dimorphism related to IRI and IRI-associate ALF. On the other hand, preclinical studies conducted on rodents with trauma-hemorrhage showed that low levels of testosterone could be beneficial for cardiovascular and hepatocellular functions in males, whereas high levels of estradiol and prolactin could improve cardiovascular dysfunctions in estrous females [65]. Data from another experimental study demonstrated that female mice were more protected than males from the negative effects of IRI on the liver due to the activity of ovarian estradiol. The detailed role played by estrogens in the amelioration of IRI-associated liver dysfunctions remains to be understood completely. Some authors suggested that estrogens enhance the production of nitric oxide (NO) in the endothelial cells of vascular endothelium, thus inducing vasodilation and enhancing the perfusion of the remnant hepatic tissue, but this hypothesis is still speculative, and has not been fully demonstrated [66].

#### 3.2.3. Hepatitis A

The massive necrosis of hepatic tissue occurring in ALF may also be due to viral infections such as Hepatitis A virus (HAV) [67]. Although the quality of food and hygiene has improved in many countries during the last few years, the risk of HAV infection is still present [68]. HAV is one of the most important causes of ALF, especially in children, but the underlying mechanisms that lead to hepatic failure are under investigation [67,69]. The incidence of HAV resulting in ALF is less than 1% of all HAV cases, wherein 69% of these patients have demonstrated a spontaneous survival rate, while the remaining 31% do not survive or require an urgent liver transplantation [69]. Furthermore, age and sex are risk factors for HAV infection, as young males are more susceptible to HAV than females, but HAV-associated ALF has not been correlated to any sex dimorphism. A retrospective study from Changa and colleagues demonstrated that patients ≥ 40 years old were more susceptible to the development of ALF than younger patients, without sex-related differences [68].

## 4. Chronic Liver Injury

Chronic liver diseases comprise a wide spectrum of pathologies with variable etiologies (including viral hepatitis, steatosis, and autoimmune liver diseases) and clinical manifestations. Irrespective of the etiology, chronic liver damage could evolve in fibrosis, cirrhosis, and hepatocellular carcinoma. Sex differences have been observed in the development and the clinical outcome of these diseases. The main recent findings regarding the sex-dependent mechanisms involved in chronic liver injury of different origins are described in this review. 

### 4.1. Viral Hepatitis

Viral hepatitis, including hepatitis B virus (HBV) and hepatitis C virus (HCV) infections, is an important cause of both acute and chronic liver damage. Despite the recent advances in HBV vaccination programs and effective HCV eradication treatments, virus-related chronic liver diseases remain a major health problem. Both chronic hepatitis B (CHB) and chronic hepatitis C (CHC) display a variable prevalence, depending on the geographical area and cultural, behavioral, and social factors influencing exposure risk, the timing of the first diagnosis, and access to pharmacological treatment. It has been generally recognized that women are less susceptible than men to viral infections by virtue of a more efficient innate, humoral, and cell-mediated immune response [70,71]. Indeed, the expression of toll-like receptors (TLRs) and the amounts of monocytes, macrophages and dendritic cells are higher in females than males in both rodents and humans, and this leads to more intense inflammatory responses. Moreover, it has been reported that the activation of antigen presenting cells (APCs) as a consequence of viral infections is generally greater in females than in males, as well as the activity of cytotoxic T cells, the CD4+/CD8+ ratio, and the number of CD4+T cells, inducing an improved engagement of the T cell receptor [72]. 

#### 4.1.1. HBV Hepatitis

In addition to the different immune response regulation observed in the two sexes, it should be underlined that HBV is a sex-hormone-responsive virus, the host infection capability of which is differently regulated by androgens and estrogens. Indeed, after its activation, the androgen receptor (AR) binds viral androgen-responsive elements in the HBV enhancer 1 region, increasing its mRNA production [73]. On the other hand, estrogen binding to its receptor ERα can directly suppress HBV enhancer 1 activity and prevent its interaction with hepatocyte nuclear factor 4α (HNF-4α), thus decreasing viral transcription [74,75]. Moreover, the greater viral response observed in females has been explained by the fact that the majority of genes determining this response are located on the X chromosome [72]. A dedicated discussion should be carried out for HBV infection in pregnancy. Even though HBV infection seems not to influence fertility, morbidity, and maternal/fetal mortality [76], particular care should be taken in pregnant cirrhotic women, who have a higher risk of developing gestational hypertension, placenta detachment, peripartum hemorrhages and caesarean delivery with respect to healthy women. Furthermore, their newborns have an increased risk of low weight at birth and preterm birth [77,78].

Another observed sex-related difference was the clinical outcome of HBV hepatitis. It has been observed that HBV-related HCC is more frequent in men than in women, with a ratio of 5–7:1 [79], which seems to be directly correlated to the higher viral response observed in women. It has also been hypothesized that HBV-induced HCC is raised during the first stage of active HBV replication in chronic HBV patients, as higher serum viral loads were causally correlated to an increased risk of developing the tumor. Therefore, women can be protected by their viral response, which is higher than that of men [74,80].

#### 4.1.2. HCV Hepatitis

In female HCV patients, a higher rate of symptoms has been reported, together with a spontaneous viral clearance when a genetic polymorphism of interleukin-28B (IL-28B)—the IL28B CC—is present, thus suggesting a synergistic effect of this genotype and female sex in favoring spontaneous viral clearance [81,82]. It has been hypothesized that this effect could be linked to the stimulation of TLR7, a receptor with a pivotal role in viral recognition and the activation of the immune response by the release of interferons (INF-α and INF-λ) [83]. Furthermore, the transcription of some antiviral and pro-inflammatory genes is under the control of estrogen-response elements, which are overexpressed in women [84], suggesting that the observed immunological sexual dimorphism could be related to the different amount of steroid hormones in the sexes. The expression of sex hormone receptors has been detected in monocytes, lymphocytes, and dendritic cells, and their activation after hormone binding triggers the release of cytokines and chemokines, modulating their differentiation, maturation, and proliferation [85]. In particular, androgens display an anti-inflammatory role, as they could increase the production of IL-10 and transforming growth factor β (TGF-β), and could inhibit IFN-γ, IL-4, and IL-5 secretion from T cells [72]. Progesterone acts similarly to androgens, by suppressing the Th1 response, activating Th2-mediated cytokine production, inhibiting cytotoxic T cells, and modulating the function of natural killer (NK) cells [70]. Estrogens could exert the opposite effects, depending on their concentration. At low concentrations, they promote monocyte differentiation into DCs, with the consequent release of IL-4 and IFN-α, Th1 cell activation, and prompt cell-mediated immune responses. On the other hand, high doses of estrogens inhibit both innate and pro-inflammatory responses, whilst activating Th2-type and humoral immune responses [86,87]. 

### 4.2. Alcoholic Liver Disease (ALD)

One of the main causes of chronic liver disease worldwide is alcoholic liver disease (ALD), accounting for about 48% of cirrhosis-related deaths in the US [88]. Alcohol consumption could also represent an additive risk factor for the development and progression of other chronic liver diseases, e.g., viral hepatitis. ALD could evolve in a broad spectrum of hepatic pathological manifestations, ranging from simple alcohol-induced steatosis to a more severe fibrotic and cirrhotic state, characterized by complications such as ascites, portal hypertension, bleeding and encephalopathy [88]. One of the first things to consider regarding the sex difference in alcoholic-related liver disease is that males are more prone than females to alcohol consumption (68% vs. 64%), and more males than females are diagnosed with an alcohol use disorder (7% vs. 4%) [89]. Moreover, females have an higher risk of developing alcohol-related liver disease with respect to males; in particular, a Danish study observed that an increased rate of alcohol-related liver disease was obtained in women consuming 7–13 beverages per week (84–156 g of relative alcohol intake) and in men consuming 14–27 beverages per week (168–324 g of relative alcohol intake) [90]. 

An increased risk of ALF after excessive alcohol consumption has been shown in females compared to males, due to the different metabolism of alcohol in the two sexes that makes women most prone to alcohol-induced hepatotoxicity [91]. This effect is due to a different genetic pattern leading to sex-influenced alcohol pharmacokinetics. This is due to differences in metabolism because of the sex-related differences in the expression of gastric and hepatic alcohol dehydrogenase, and because of different gastric absorption due to changes in gut permeability [91,92].

The sex-related differences in alcohol consumption and in the risk of developing alcohol-induced liver disease were also observed in the progression of hepatic damage, as fibrosis progresses more rapidly in women than in men, and persists even after alcohol abstinence [93]. Several factors have been proposed to explain this increased ALD prevalence and severity in women. Some researchers have found that chronic alcohol ingestion dramatically affects the blood and liver expression of hormones in both sexes, thus modulating hormone-controlled pathways [94,95]. The estrogen-induced activation of Kupffer cells following alcohol administration in female rats caused hepatocyte inflammation and necrosis, which sustain damage progression. Moreover, the administration of an estrogen antagonist led to the reduction of both inflammation and necrosis [96]. In female rats, ER expression was not affected by alcohol administration, whereas in male rats, alcohol induced an increase of ER expression, as well as an increased hepatocyte proliferation rate [97]. Moreover, it has been demonstrated in animal models that excessive exposure to alcohol decreases the expression of many hepatoprotective genes in females, along with some genes involved in compensatory pathways, whereas inflammation and oxidative stress are increased in males by alcohol consumption [98,99,100]. Another factor involved in sex-related ALD development is related to the different modulation of alcohol-induced endotoxemia in the two sexes. It has been demonstrated that an LPS increase leads to increased intestinal permeability and stimulates the release of pro-inflammatory mediators from Kupffer cells, and ALD patients showed a more severe “leaky gut” with respect to patients without ALD [101,102]. A study by Kirpich and collaborators observed an increased level of some biomarkers, such as ALT and AST levels, CK18 M65 and CK18 M30, and reduced levels of flagellin in ALD females compared to males, suggesting that females are more susceptible to ALD than males, probably because of a different adaptation to chronic alcohol-induced changes in gut permeability [103]. Further studies are needed in order to understand the correlation between the differences in gut microbiome and function between males and females, and the more rapid progression of ALD in females. 

### 4.3. Non-Alcoholic Liver Disease (NAFLD)

Non-alcoholic fatty liver disease (NAFLD) is a widespread liver disease characterized by the accumulation of fatty acids in hepatocytes. This condition ranges from simple fat deposition in the hepatic parenchyma to lipotoxicity and inflammatory damage, leading to the development of non-alcoholic steatohepatitis (NASH). NAFLD has a high prevalence in Western countries, e.g., the US and the EU, and is considered to be the hepatic manifestation of metabolic syndrome, as it is generally associated with obesity, insulin resistance and visceral adiposity [104], and also with extrahepatic manifestations [105]. Sex-related differences in the prevalence and the clinical outcome of NAFLD/NASH have been studied, even though conflicting results have been reported [42].

The liver is involved in the regulation of energy metabolism, orchestrating the conversion and storage of glucose as glycogen, and producing fatty acids (FAs) starting from the acetyl-CoA deriving from glycolysis [106]. In the male liver, the synthesis of FAs is more efficient than that in the female liver due to sex-related differences in the circulating levels of insulin and leptin during the postprandial phase. Indeed, both in vitro and in vivo studies have demonstrated that testosterone reduces the concentration of insulin, while estrogens upregulate the production of leptin, which in turn inhibits some insulin-mediated pathways. Sex differences in hepatic *Fasn* expression have been attributed to elevated levels of insulin in males, whereas higher rates of leptin in females were responsible for blocking the activity of insulin, thus resulting in a decreased gene expression of *Fasn* and lipid metabolism [107]. In females, FAs are cleared from plasma faster than they are in males, due to an increased transport rate because of the higher expression of several transport proteins, such as the cluster of differentiation 36 (CD36) and FA transport proteins (FATPs) [108,109]. In general, the female liver can better respond to an excessive lipid intake than the male liver, due to an improved capability of regulating the uptake of FAs and the synthesis of triglycerides (TGs) [110]. A combination of higher fat oxidation rates and a well-regulated de novo lipogenesis is responsible for a lower postprandial hepatic lipid accumulation in women than their male counterparts, thus reducing the risk of hepatic steatosis. Notably, increased triacylglycerol-rich (TAG-rich) very-low-density lipoprotein (VLDL) secretion and LPL-mediated VLDL-TAG clearance in response to excessive lipid intake and to obesogenic-like conditions were observed in females, thus maintaining lower levels of circulating VLDL-TAG [111]. Moreover, in men and postmenopausal women, higher circulating levels of low-density lipoprotein (LDL)-cholesterol and lower levels of high-density lipoprotein (HDL)-cholesterol have been observed with respect to premenopausal women, and these conditions were associated with an increased incidence of liver steatosis [112]. In ovariectomized mice, characterized by a drop in the levels of endogenous estrogens, an increase of hepatic fat deposition has been observed. This increase was reverted after estrogenic supplementation, supporting the direct role of estrogens in the prevention of hepatic lipid accumulation. In order to strengthen these findings, Meda and colleagues demonstrated that in female mice fed with a dietary excess of lipids, ERα prevented the deposition of hepatic lipids by inhibiting lipid synthesis and regulating the lipid intake, whereas in male mice fed with the same dietary supplements, ERα exerts a negative role by promoting lipid accumulation [110,113]. In addition, some findings support the protective effect of E2 on steatosis in menopause, as estrogenic supplementation in post-menopausal females with NAFLD could improve liver function, even though further studies are needed to confirm this hypothesis [114,115]. The deletion of ER𝛼 or GPER induces liver steatosis and fat accumulation in both female and male mice by acting on lipid metabolism and transport. In particular, as a consequence of the genetic deletion of estrogen receptors, the upregulation of sterol regulatory element binding protein 1c (SREBP-1c) and some lipid transport genes, and the dysregulation of diacylglycerol acyltransferase DGAT1/2 and insulin-stimulated acyl-coA carboxylase phosphorylation has been observed [9,116].

### 4.4. Autoimmune Liver Diseases: PBC, PSC, AIH

Autoimmune liver diseases are chronic conditions characterized by persistent hepatic inflammation and the activation of autoimmune responses. They comprise a range of different diseases, including primary biliary cholangitis (PBC), primary sclerosing cholangitis (PSC) and autoimmune hepatitis (AIH). Despite the recent advances in the field, the mechanisms underlying these diseases and the eventual role of sex in their onset, development and clinical history are far from being completely elucidated. 

#### 4.4.1. Primary Biliary Cholangitis

Primary biliary cholangitis (PBC) is an autoimmune disease and one of the most frequent types of chronic liver cholestatic diseases, characterized by the progressive destruction of intrahepatic bile ducts and cholangiocytes, leading to inflammation and cirrhosis [42,117]. In most cases (20–60%), the diagnosis of PBC occurs in the absence of symptoms by the analysis of biochemical markers, such as serum levels of alkaline phosphatase (ALP) and anti-mitochondrial antibodies (AMAs) [118]. Most symptomatic patients are middle-aged women with manifestations of fatigue, pruritus, and skin hyperpigmentation. As these features are commonly associated with aging, the disease incidence can be underestimated, and symptoms are frequently overlooked by patients [117]. The clinical symptoms of PBC are strictly related to sex, as an accumulation of female sex hormones during the administration of oral contraceptives and pregnancy is responsible for pruritus in women, while males tend to develop more gastrointestinal bleeding than their female counterparts [119]. The pathophysiology of PBC is predominantly associated to a loss of tolerance towards the mitochondrial E2 component of the pyruvate dehydrogenase complex (PDC-E2) on the intracellular biliary endothelial cells (BECs), or cholangiocytes [117,120]. AMAs are a specific marker of PBC, as they target only the small bile ducts, even though mitochondria are ubiquitously present [121]. This type of reaction has been directly associated with the recruitment of autoreactive T-lymphocytes, such as CD4+ and CD8+ T cells, monocytes and natural killer (NK) cells [117]. The influences of genetic variations associated to the human leukocyte antigens (HLA)-DR7 and DR-8, sex, smoking and co-morbidities are risks factor for PBC [118]. Although women are more susceptible to the development of PBC, with a 1:10 male: female ratio [120], the disease phenotype and the survival rate is the same in both sexes, with a slower progression of liver fibrosis in female patients [34,42]. According to many epidemiological studies, there are no specific male-related risk factors for the development of PBC, while many risk factors—such as estrogen deficiency—have been identified in females [119]. Indeed, the pathophysiological mechanisms of PBC could be dictated by sex, as sex steroids could alter the HLA gene, cytokine production, the proliferation of cholangiocytes, and the recruitment of some immune cells [34]. Women with PBC demonstrated a major antibody production, together with increased levels of CD4+ T cells with respect to males, and this is probably due to the role of sex hormones in the modulation of the functioning of immune cells. Notably, estrogen and androgen receptors are both expressed on B cells, while CD4+ and CD8+ T cells only express estrogen receptors [119,122]. The potential role of estrogens on the biliary tree has been studied in a bile-duct-ligated (BDL) model of ovariectomized (OVX) female rats, where a lower expression of ERβ was associated with a reduced bile duct mass. Interestingly, the exogenous administration of estradiol (E2) was able to repristinate cholangiocyte proliferation through the Src-Shc-ERK1/2 signaling pathway [123]. In order to investigate the presence of sex-related differences in AMA-mediated reactions, Nalbandian and colleagues evaluated the serological levels of AMAs and the reactivity of PDC-E2 in a cohort of 42 patients of both sexes, demonstrating the lack of significant differences between females and males in the levels of AMA, meaning that they had the same AMA reactivity [119]. However, the reason for the different PBC prevalence in the two sexes could reside in the X chromosome. Hence, similar immune-mediated pathologies, such as the X-linked severe combined immunodeficiency (XSCID) and Immunodysregulation polyendocrinopathy enteropathy X-linked (IPEX) are due to mutations on Xp11 and Xq13.1, respectively. Nevertheless, asymmetric X chromosomes lead to other autoimmune diseases that are concomitant with PBC, including autoimmune thyroidal pathologies and sclerosis. In addition, epigenetic modifications—such as the methylation of some X-regulated genes, X chromosome monosomy, and the inactivation of X chromosomes—could be associated with the progression of PBC [34].

Besides all of the hypotheses regarding the female prevalence of PBC, the pathophysiological mechanisms of PBC still remain elusive per se, even though the sex-related differences are mostly related to the different subset of immune cells observed in the two sexes, or to specific HLA genetic variants, suggesting that the adaptive and acquired immune response is involved in PBC pathogenesis.

#### 4.4.2. Primary Sclerosing Cholangitis

Primary sclerosing cholangitis (PSC) is a rare, idiopathic hepatobiliary chronic condition in which the inflammation of both intra- and extra-hepatic bile ducts leads to a progressive “onion skin” fibrosis within the hepatic parenchyma and in the biliary tree [124]. This pathology mostly affects men, with a 2:1 male : female ratio, and is more frequent in young and middle-aged men affected by inflammatory bowel disease (IBD) [125]. The diagnosis in women tends to take place later than in men, approximately at 45 years of age [120]. In total, 50% of patients are asymptomatic and are diagnosed accidentally. The first symptoms, such as pruritus, weight loss and abdominal pain, arise during the late stages of the disease. In order to confirm the diagnosis of PSC, biochemical analysis for the determination of bilirubin, serum aspartate and alanine aminotransferase should be performed. Interestingly, PSC could not be considered a classical autoimmune disorder like PBC [42], as the titers of anti-mitochondrial and antinuclear antibodies have been found to be lower in PSC patients than in those with PBC, also supporting the hypothesis of a different etiology [120,125]. To date, PSC pathophysiology is far from fully understood, and this contributes to the lack of effective pharmacological treatment. The only clinical intervention is still liver transplantation [126].

Because of inflammation, immune cells and fibrosis accumulate around biliary tracts and portal areas, affecting the cholangiocytes lining the intra- and extrahepatic bile ducts [127]. A single-center study identified some sex-dependent differences in PSC patients. For example, it has been observed that coffee consumption might be protective from PSC development in male patients through the enhancement of the bile flow, regardless of the presence of co-morbidities. On the other hand, coffee’s positive effects have not been observed in women [128]. Andersen and colleagues proposed a correlation between female reproductive health and PSC. From this single-center study, the results suggested that there was a linear correlation between parity and women’s age at the time of the PSC diagnosis, suggesting that childbearing might delay the development of this disease [128]. However, this hypostasis seems not to be supported by other clinical evidence, as no correlation has been demonstrated between the age of PSC diagnosis and the number of children. Moreover, the most recent and largest single-center study from Wronka and colleagues confirmed that PSC did not affect pregnancy and delivery [129]. 

As the main pathophysiological mechanisms involved in PSC development are still lacking, only a little evidence has been reported to explain the male prevalence of this disease, mostly indicating a genetically related individual susceptibility that could be accompanied by some environmental factors causing immunomediated cholangiocyte damage, e.g., vitamin D and selenium deficiency [130].

#### 4.4.3. Autoimmune Hepatitis

Autoimmune liver diseases include autoimmune hepatitis (AIH), with a small percentage (~22%) of patients manifesting acute injuries typical of ALF evolving in chronic disease [33]. AIH is characterized by injuries to the hepatic parenchyma, leading to progressive inflammation and cell disruption [42]. Its pathophysiology has been associated with damage within the liver tissue due to the impairment of regulatory T cells [72]. The exact triggering pathways involved in AIH are under investigation, but there is a consensus that AIH is characterized by a loss of tolerance towards hepatocytes. Presumably, the epitope of liver autoantigens can bind to a particular region of HLA II, and is subsequentially expressed on APC cells. Antigenic epitopes of toxins and exogenous substances are able to trigger immune cells, including CD4+, CD8+ T, NK and B cells, by reacting with specific hepatocyte antigens [131]. AIH epidemiology and incidence correlated to genetic variations of Major Histocompatibility Complex (MHC) class II genes, in particular the HLA-DR locus. European and North America Caucasian patients with HLA-DRB1*0301, DRB3*0101 and DRB1*0401 alleles have a higher probability of developing AIH. On the other hand, the HLADRB1*1302 allele exerts a protective effect in Argentinians and Brazilians [132]. Furthermore, the AIH-associated specific haplotype HLA A1-B8-DR3 is more present in AIH male patients than in women [72]. Although no clear age- or sex-related differences in the onset age and clinical signs of the disease could be identified, the prevalence of AIH is higher in females, with a female: male ratio of 3.6:1 [72,131]. To date, prednisolone and azathioprine are the only effective therapy for AIH. Controversial results regarding an influence of sex in the responses to the treatment have been reported [42]. For instance, Czaja and colleagues did not identify sex-related responses to steroids, while Al-chalabi and collaborators found a higher survival rate among steroid-treated males with respect to females [42,133]. In addition, it has been observed that the excretion of estrogens during the second semester of pregnancy could alleviate the severity of AIH, probably due to the activation of an anti-inflammatory response [72]. Indeed, it has been hypothesized that sex hormones might have a role in the modulation of the immune system after their linkage to their receptors on the immune cells, thus managing the antigen presentation, the release of cytokines, and other mechanisms [42].

### 4.5. Liver Cancers

Liver tumors comprise a wide range of benign and malignant manifestations affecting different hepatic cell types, ranging from benign adenomas to hepatocellular carcinoma or cholangiocarcinoma. Regarding benign tumors—e.g., cavernous hemangioma, focal nodular hyperplasia (FNH), hepatic adenoma, biliary cystadenoma, and solitary hepatic cysts—a higher prevalence in females could be demonstrated, which has been linked to estrogen levels. However, a clear causal mechanism has not been demonstrated, nor has a correlation with the assumption of exogenous estrogens with oral contraception. This female-prone prevalence appears to be inverted in malignant tumors, the prevalence of which is higher in males [34]. Men are two to three times more prone than women to the development of hepatocellular carcinoma (HCC)—the most common primary hepatic cancer, with 80% of all cases—due to a male-specific cluster of risk factors, and also different sexual hormone expression [134]. Some reports indicate a male-to-female incidence ratio of 3–4:1 for human liver cancer in general, even after normalization for differences in risk factors [34,135]. Similarly, intrahepatic cholangiocarcinoma is more common in males than in females [136]. A sex difference in hepatic tumor development has also been observed in different animal models of liver cancer, both transgenic [137] and obtained by the administration of a toxic compound, such as 4-aminobiphenyl (ABP) [138], AFB1 [139], or diethylnitrosamine (DEN) [140].

As HCC is the life-threatening outcome of many chronic liver diseases, the sex-related disparity observed in HCC development could be the consequence of the sex differences observed in hepatic diseases. The pathophysiology underlying the sex difference in HCC has been related to the fact that it is an androgen-sensitive tumor, and sex hormones are able to promote cancer cell proliferation. IL6 seems to play a central role in sex-related HCC progression, as IL6 knockdown abolishes sex-related hepatocarcinogenesis in mice. Indeed, IL6 is a target of FOXA, a transcription factor regulating estrogen and androgen signaling [141], and estrogens are known to decrease IL6 production, thus reducing HCC progression [142,143]. A study demonstrated that estrogens counteract IL6 secretion from liver Kupffer cells, resulting in a decrease of tumor growth in mice [142]. 

Another study demonstrated that IL6 transcription could be upregulated by the binding of nuclear factor erythroid 2-related factor 2 (NRF2) to the antioxidant response elements within its promoter [144]. This factor is involved in the orchestration of immune responses to infection, maintaining redox balance and inflammatory signaling cascades, and in protecting the liver against carcinogen-induced tumorigenesis. In APB-treated female mice, a stronger NRF2-associated antioxidant response was observed with respect to males, as well as a reduced CYP2E1-related ABP-N-oxidization, leading to a female-specific reduction of oxidative stress and an increase of antioxidant defense [145]. 

Several studies have investigated the efficacy of anti-estrogen therapy with tamoxifen in patients with advanced HCC, as approximately one third of HCC patients display ER expression, and an estrogen-dependent HCC growth has been observed. Nevertheless, these studies failed to observe an improvement in survival or liver functional status [146,147,148]. Furthermore, studies investigating antiprogestin and antiandrogen agents failed to demonstrate efficacy in HCC therapy [142,149]. 

Another aspect involved in sex differences related to liver cancer is linked to the above-mentioned sexual dimorphism of the immune system, with a higher number of innate immune cells—e.g., monocytes, macrophages and dendritic cells—and adaptive immune CD4+ T cells in females with respect to males, who in turn have a higher number of CD8+ T cells [150]. Moreover, a tenfold increase of TLRs—a class of protein named pattern recognition receptors (PRRs) involved in regulation of innate immune response—was observed in females [151]. The sexual dimorphism of the immune system is reflected in the different sex-related susceptibility to anticancer immunotherapy that has been observed in many cancer types [152,153]. Interestingly, it has been observed that sex hormones control the gene transcription of both *PD1* and *PD-L1*, which are checkpoint proteins that are also upregulated in many hepatic tumors [154,155,156], and that males seem to respond better than females to therapy with checkpoint inhibitors [157]. 

Even though the evidence regarding the different response to checkpoint blockade between the two sexes remains conflicting and far from fully understood, the reasons are likely to be related to the higher tumor mutation burden in males than in females [150,158]. Indeed, a higher tumor mutation burden leads to better tumor immune recognition after immunotherapy. Other proposed reasons for the sex-related difference in response to immunotherapy are the presence of epigenetic factors, e.g., tobacco smoking, and the fact that the increase of immune response after immunotherapy is less effective in women than in men because women exhibit per se a stronger immune environment than men [150,158].

Table 1 summarizes the main sex-related differences in acute and chronic liver diseases that have been described in detail in the text. 

## 5. Conclusions

Because the liver is a highly sexually dimorphic organ, accounting for at least 72% of sexually differentiated genes, the pathophysiological mechanisms involved in many acute and chronic liver diseases have been demonstrated to be differently regulated in females and males. In recent years, many efforts have been devoted to understanding the effect of sex on the development and progression of liver diseases. Accordingly, a growing amount of epidemiological, clinical, and experimental data shows significant differences in the development, progression and clinical outcome of conditions that are common to men and women, adverse drug reactions, the efficacy/effects of such treatments and nutrients, and lifestyles. Regarding the clinical point of view, despite increasing evidence that an individual’s sex is one of the most important modulators of disease risk and response to treatment, consideration of the patient’s sex in clinical decision making (including the choice of diagnostic tests, medications, and other treatments) is often lacking.

Being still far from the definition of a liver gender medicine, this review summarized the main sex-related differences which have been highlighted so far in the development and severity of liver dysfunctions of different etiologies. It is of note that these differences are still descriptive or mechanistic, and have not determined attempts at personalizing the therapeutic interventions according to the patients’ gender. Therefore, the need of developing new personalized pharmacological treatments which are able to better tailor the therapy according to sex differences remains to be pursued. 

## Figures and Tables

**Figure 1 ijms-23-10654-f001:**
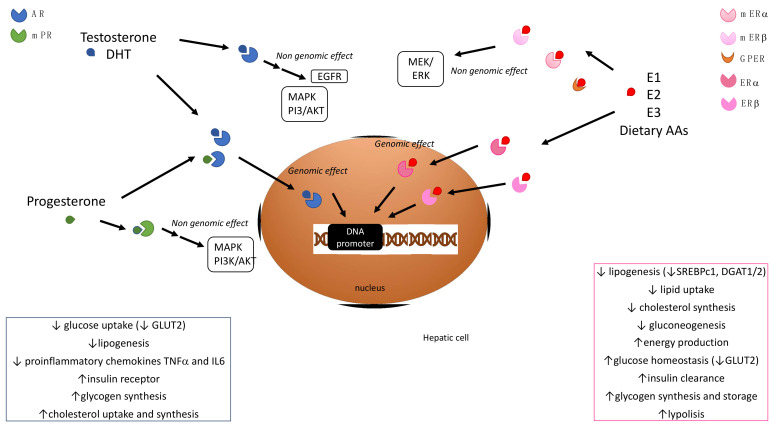
Main signaling pathways mediated by sex hormone receptors in hepatocytes and their physiological effects. AR, androgen receptor; mPR, membrane progesterone receptor; DHT, dihydrotestosterone; MAPK, mitogen-activated protein kinase; PI3K/AKT, phosphatidylinositol 3-kinase/protein kinase B; EGFR, epidermal growth factor receptor; MEK/ERK, mitogen-activated protein kinase kinase/extracellular signal-regulated kinase; E1, estrone; E2, estradiol; E3, estriol; mER, membrane estrogen receptor; ER, estrogen receptor; GPER G, protein-coupled estrogen receptor 1; AA, amino acid; GLUT2, glucose transporter 2; TNFα, tumor necrosis factor α; IL6, interleukin 6; DGAT, diacylglycerol acyltransferase; SREBPc1, sterol regulatory element binding protein 1c.

**Table 1 ijms-23-10654-t001:** Relative incidence and male : female ratio and sex difference in acute and chronic liver diseases. N.a., not available; DILI, drug-induced liver injury; CYP, cytochrome P450; P-gp, P-glycoprotein; NAFLD, non-alcoholic liver disease; PBC, primary biliary cholangitis; PSC, primary sclerosing cholangitis; AIH, autoimmune hepatitis; HCC, hepatocellular carcinoma; TLR, toll-like receptor; PD1, programmed cell death protein 1; PDL1, Programmed death-ligand 1; FA, fatty acid; HLA, human leukocyte antigen; E2, estradiol.

Liver Disease	Relative Incidence Male:Female Ratio	Mechanisms of Sex Differences	Refs.
**Acute liver injury**			
DILI (according to RUCAM)	1:2	Sex-related different bioavailability and excretion of drugs e.g., due to sex hormone activity that affect CYP and P-gp expression Difference in genetic backgrounds	[47,49,51,52]
**Chronic liver disease**			
Viral hepatitis	Conflicting results, Female generally have higher rate of symptoms but increased viral clearance	Females display more efficient innate, humoral and cell-mediated immune response (higher cytotoxic T cells, higher CD4+/CD8+ ratio and higher CD4+ T cells), as well as more TLRs	[70,71,72,84]
ALD	1:2	Estrogen-induced activation of KCs after alcohol administration in female rats increases hepatocyte inflammation and necrosis Alcohol exposure in female decreases upregulation of hepatoprotective genes, and genes involved in compensatory pathways, inflammation and oxidative stress	[98,99,100,103,159]
NAFLD	n.a.	Higher FA clearance and synthesis in females (increased FA transport protein expression) Higher LDL-cholesterol in men and postmenopausal women E2 seems to be protective for NAFLD	[107,110,112,114,115]
PBC	1:10	Estrogen-dependent alteration of HLA expression, cytokine release and cholangiocyte proliferation	[34,120,123]
PSC	2.6:1	Few evidences suggest a correlation between a good female reproductive health and childbearing and delay of PSC development	[128,129]
AIH	1:3.5	Female hormone-related modulation of immune system improving AIH-induced inflammation	[42,72]
Benign hepatic cancerous lesions	1:5–15 (depending on types)	Estrogens improve the outcome of benign lesions	[160,161]
HCC	3–4:1	Estrogen-modulated IL6 decrease in females improves HCC progression, through regulating NRF2-antioxidant response Increased expression of TLRs involved in the innate immune response, Higher rate of CD4+ cells in females with respect to males Males better respond than female to checkpoint blockade therapy since sex hormones control PD1-PDL1 expression.	[144,145,150,151,154,155,156,157]

## Data Availability

Not applicable.

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
