# Peer review of "The Role of Sex in Acute and Chronic Liver Damage"

_ijms, 2022, doi:10.3390/ijms231810654_

Round 1

Reviewer 1 Report

Comments are in the attached file

Author Response

The Review Article IJMS 1885762 entitled "The role of sex in acute and chronic liver disease” by Katia Sayaf, Daniela Gabbia, Francesco Paolo Russo and Sara De Martin describes how sex influences the outcome of liver disease. The first part of the review focus on sex hormones and their implication in liver metabolism while the second one is centered in the role of sex hormones in different liver diseases.

The manuscript is well organized and will be of interest for the scientific community after minor revision of some parts of the text. In general, most of the diseases described in the manuscript are well addressed, however I feel some parts are inconclusive regarding the impact of sex differences in the disease, mainly in reference to the autoimmune liver diseases such as primary biliary cholangitis and primary sclerosing cholangitis. Further comments on this aspect will be interesting.

We thank the reviewer for the meticulous review and discussion. We improved the conclusion of the subsections 4.3.1 and 4.3.2 regarding the mechanisms of sex-related differences in the pathophysiology of PBC and PSC.

On page 10 the last sentence is too long (lanes 488-496) and it makes not sense. Rewrite and if possible divide into shorter sentences in order to facilitate its understanding and relevance.

We thank the reviewer for its suggestion and modified the sentence accordingly.

My last comment is concerning the Conclusion section of the manuscript. In my opinion this section is not proportional to the information presented in the main text. A “take-home” message would be appropriate to be incorporated as well as additional prospects, trouble shootings and even some degree of speculation about the role of sex hormones.

We thank the Reviewer for this suggestion and add some new information throughout the text and in the conclusion section.

Other minor comments that I think the authors should address are as follows:

- Abbreviations should be carefully revised:

  • Some of them are missing:

P-gp (lane 192 pag 5)
IL6 (lane 205 pag 5
HBV and HCV (lane 221 pag 5) DCs (lane 253 pag 6)
TGFb (lane 249 pag 6)
TG (lane 289 pag 7)
TAG (lane 292 pag 7)
VLDV (lane 293 pag 7)
LDL (lane 293 pag 7)
HDL (lane 296 pag 7)
BDL (lane355 pag 8)

We added the acronym explanation in the text.

  • They must be explained the first time they appear:

DILI (lane 141 pag 4)
HLA (lane 164 pag 4)
APC (lane 177 pag 4)

We modified the text accordingly.

Eliminate those citations only cited once and those with different meaning (ARE stands for androgen responsive elements and antioxidant response elements)

We deleted the acronyms with different meanings and those only cited one time.

-  Revise and maintain spelling of T cell (CD4+T or CD4 T)

We uniformed all to CD4+ T.

-  Some term are not explained and could make difficult to understand: logPs (lane 199 pag 5)
IL28B CC genotype (lane238 pag 6)

We modified accordingly.

In general, the whole text should be carefully revised to eliminate unnecessary uppercase lettering and redundancy of abbreviations explanation.

We thank the reviewer for the careful check of the text and revised the manuscript accordingly.

Reviewer 2 Report

Journal: IJMS                                                                          
Manuscript ID: ijms-1885762

Authors: Katia Sayaf et al.

Title: “The Role of Sex in Acute and Chronic Liver Damage”

The authors of the present manuscript review the available data regarding the role of sex in acute and chronic liver damage and the potential underlying mechanisms. It is an interesting topic. Below there are some comments and suggestions for consideration.

Comments:

1.     I would suggest mentioning the aim(s) of the review at the end of the introduction section.

2.     Please define in figure1 the genomic and non-genomic effects of estrogens. In addition, please define all abbreviations used in the figure in the figure legend.

3.     Likewise, please define all the abbreviations used in the table. Please add the relevant references.

4.     I would suggest creating different paragraphs for chronic hepatitis B and chronic hepatitis C.

5.     For the statements: “The role of sex in influencing the pathophysiological mechanisms and outcomes of ALF is far to be completely understood. However, a role of sex hormones has been proposed, as well as sex-related modifications of gene transcription and consequent metabolic disorders.” and “Generally, about the 67%-70% of ALF patients are women, especially in the case of DILI, mirroring the typical susceptibility of women to adverse drug reactions (ADRs).”, please add the relevant references.

6.     In the 3rd section of the review (“Acute liver injury”), only Drug-induced liver injury (DILI) is discussed in more detail. Despite that data might be limited for some cases of ALF in terms of completeness, it would be important if the authors could summarize the current evidence or the limited available data (potential sex-dependent pathophysiological mechanisms, epidemiological data, etc.) for these causes (e.g., ischemia, ischemic reperfusion injury, etc.). If no data/evidence of sex-dependent associations exists for some causes, please clarify this in the relevant section as well.

7.     Please also discuss the sex differences regarding alcoholic liver disease / alcohol-induced liver injury.  

8.     Please briefly summarize the clinical implications derived from this review and propose future directions in the field of research before the conclusion.

Author Response

Reviewer 2:

The authors of the present manuscript review the available data regarding the role of sex in acute and chronic liver damage and the potential underlying mechanisms. It is an interesting topic. Below there are some comments and suggestions for consideration.

Comments:

  1. I would suggest mentioning the aim(s) of the review at the end of the introduction section.

We thank the reviewer for the revision and added the aim as suggested.

  1. Please define in figure1 the genomic and non-genomic effects of estrogens. In addition, please define all abbreviations used in the figure in the figure legend.

We thank the reviewer for the suggestion and modified the figure and its legend accordingly.

  1. Likewise, please define all the abbreviations used in the table. Please add the relevant references.

We added the abbreviation and the references in the table.

  1. I would suggest creating different paragraphs for chronic hepatitis B and chronic hepatitis C.

We separate HBV and HCV into two different paragraphs as suggested by the reviewer.

  1. For the statements: “The role of sex in influencing the pathophysiological mechanisms and outcomes of ALF is far to be completely understood. However, a role of sex hormones has been proposed, as well as sex-related modifications of gene transcription and consequent metabolic disorders.” and “Generally, about the 67%-70% of ALF patients are women, especially in the case of DILI, mirroring the typical susceptibility of women to adverse drug reactions (ADRs).”, please add the relevant references.

       The proper references were added in the text.

  1. In the 3rd section of the review (“Acute liver injury”), only Drug-induced liver injury (DILI) is discussed in more detail. Despite that data might be limited for some cases of ALF in terms of completeness, it would be important if the authors could summarize the current evidence or the limited available data (potential sex-dependent pathophysiological mechanisms, epidemiological data, etc.) for these causes (e.g., ischemia, ischemic reperfusion injury, etc.). If no data/evidence of sex-dependent associations exists for some causes, please clarify this in the relevant section as well.

We improved the section including the data on ALF caused by ischemia, ischemic reperfusion injury, as suggested by the reviewer.

  1. Please also discuss the sex differences regarding alcoholic liver disease / alcohol-induced liver injury.  

We thank the reviewer for the suggestion, and we added a new section regarding alcoholic liver disease

  1. Please briefly summarize the clinical implications derived from this review and propose future directions in the field of research before the conclusion.

We thank the reviewer for the suggestion, we added our comments in the final discussion

Round 2

Reviewer 2 Report

Journal: IJMS                                                                           
Manuscript ID: ijms-1885762(Revised version)

Authors: Katia Sayaf et al.

Title: “The Role of Sex in Acute and Chronic Liver Damage”

The authors of the present review article have satisfactorily responded to my comments and suggestions. They have also made the necessary changes to the paper. As a result, the revised manuscript has been significantly improved. It is an interesting paper focusing on an important research field. Therefore, there are no further comments.